# TAP-Vid: A Benchmark for Tracking Any Point in a Video

**Carl Doersch**[*]    **Ankush Gupta**[*]    **Larisa Markeeva**[*]    **Adrià Recasens**[*]
**Lucas Smaira**[*]    **Yusuf Aytar**[*]    **João Carreira**[*]    **Andrew Zisserman**[*†]    **Yi Yang**[*]

[*]DeepMind    [†]VGG, Department of Engineering Science, University of Oxford

## Abstract

Generic motion understanding from video involves not only tracking objects, but also perceiving how their *surfaces* deform and move. This information is useful to make inferences about 3D shape, physical properties and object interactions. While the problem of tracking arbitrary physical points on surfaces over longer video clips has received some attention, no dataset or benchmark for evaluation existed, until now. In this paper, we first formalize the problem, naming it *tracking any point (TAP)*. We introduce a companion benchmark, *TAP-Vid*, which is composed of both real-world videos with accurate human annotations of point tracks, and synthetic videos with perfect ground-truth point tracks. Central to the construction of our benchmark is a novel semi-automatic crowdsourced pipeline which uses optical flow estimates to compensate for easier, short-term motion like camera shake, allowing annotators to focus on harder sections of video. We validate our pipeline on synthetic data and propose a simple end-to-end point tracking model *TAP-Net*, showing that it outperforms all prior methods on our benchmark when trained on synthetic data. Code and data are available at the following URL: https://github.com/deepmind/tapnet.

## 1 Introduction

Motion is essential for scene understanding: perceiving how people and objects move, accelerate, turn, and deform improves understanding physical properties, taking actions, and inferring intentions. It becomes even more crucial for embodied agents (i.e. robots), as many tasks require precise spatial control of objects over time. While substantial attention has been given to motion estimation, e.g., through algorithms for correspondence and tracking, this paper addresses one particularly under-studied problem: long-term motion estimation of points on generic physical surfaces in real scenes.

There is a multitude of tracking algorithms which address this problem, but partially and with different weaknesses, illustrated in Figure 2. Popular box and segment tracking algorithms provide limited information about deformation and rotation of surfaces. Optical flow can track any point on any surface, but only over pairs of frames, with limited ability to estimate occlusion. Keypoint matching algorithms common in structure from motion typically detect sparse interest points and are not designed for deformable and weakly-textured objects. For some of the most common tracking problems (e.g. faces, hands, and human poses), researchers thus build trackers for hand-chosen semantic keypoints, which is extremely useful but not scalable to arbitrary objects and surfaces.

Our work aims to formalize the problem of long-term physical point tracking directly. Our formulation, *Tracking Any Point* (TAP) is shown in Figure 1. We require only that the target point and surface can be determined unambiguously from a single pixel *query* (i.e. we don't deal with transparent objects, liquids, or gasses), and directly estimate how it moves across a video. Our task can be seen as an extension of optical flow to longer timeframes with occlusion estimation, or an extension of

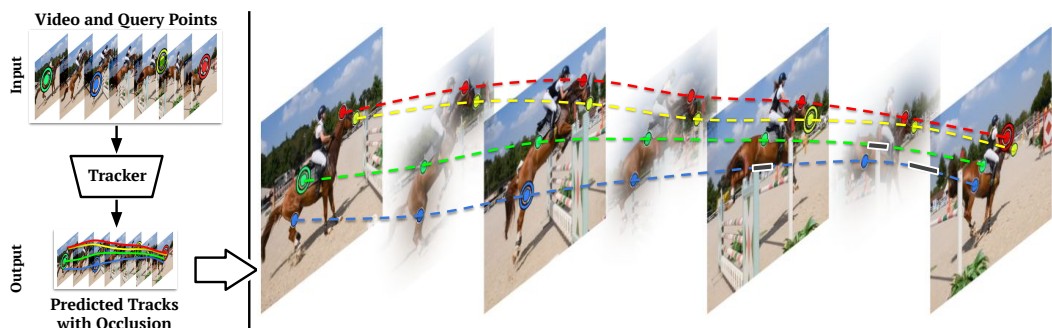

Figure 1: **The problem of tracking any point (TAP) in a video.** The input is a video clip (e.g. 10s long) and a set of query points ($x, y, t$ in the pixel/frame coordinates; shown with double circles). The goal is to predict trajectories ($x, y$ pixel coordinates; coloured lines) over the whole video, indicating the same physical point on the same surface, as well as a binary occlusion indicator (black solid segments) indicating frames where it isn't visible.

typical structure-from-motion keypoint matching to typical real-world videos, including e.g. nonrigid or weakly-textured objects and surfaces, also with occlusion estimation. To train and evaluate, we rely on a mixture of real benchmarks (where humans can directly perceive the correctness of point tracking) and synthetic benchmarks (where even textureless points can be tracked).

While tracking synthetic points is straightforward, obtaining groundtruth labels for arbitrary real-world videos is not. Thankfully, there's good evidence that humans (and other animals) excel at perceiving whether point tracking is accurate, as it is an example of the "common fate" Gestalt principle [17, 61, 65, 68, 69]. However, *annotating* point tracks in real videos is extremely time-consuming, since both objects and cameras tend to move in complex, non-linear ways; this may explain why this problem has received so little attention in computer vision. Therefore, in this work, we first build a pipeline to enable efficient and accurate annotation of point tracks on real videos. We then use this pipeline to label $1,189$ real YouTube videos from Kinetics [7] and 30 DAVIS evaluation videos [49] with roughly 25 points each, using a small pool of skilled annotators and multiple rounds of checking to ensure the accuracy of the annotations. Overall we find that on average, roughly 3.3 annotator hours are required to track 30 points through every frame on a 10-second video clip.

The contributions of this paper are threefold. First, we design and validate an algorithm which assists annotators to more accurately track points. Second, we build an evaluation dataset with 31,951 (31,301+650) points tracked across 1,219 (1,189 + 30) real videos. Third, we explore several baseline algorithms, and we compare our point tracking dataset to the closest existing point tracking dataset—JHMDB human keypoint tracking [29]—demonstrating that training using our formulation of the problem can increase performance on this far more limited dataset.

## 2    Related Work

Though box and segment tracking has been widely benchmarked [11, 24, 47, 49, 77, 79, 82, 84], we focus here on methods that track points. A few early hand-engineered works address long-term tracking of surface points [4, 38, 39, 53, 58, 76], though without learning, these approaches are brittle. **Optical Flow** estimates dense motion, but only between image pairs [22, 40]; modern approaches [13, 25, 51, 63, 66] train and evaluate on synthetic scenes [6, 13, 46, 62], though depth scanners provide limited real data [16]. **Structure-From-Motion (SFM)** [12, 30, 43, 48] relies on sparse keypoint matches given image pairs of rigid scenes, and typically ignores occlusion, as it can use geometry to filter errors [21, 55, 71, 73]. This work is thus often restricted to rigid scenes [45, 87], and is benchmarked accordingly [3, 10, 36, 36, 37, 57, 88]. **Semantic Keypoint Tracking** typically chooses a (very small) set of object categories and hand-defines a small number of keypoints to track [29, 50, 60, 64, 70, 80], though some variants track densely for humans [19, 44]. **Keypoint discovery** aims to discover keypoints and correspondence on more general objects, typically with the goal of robotic manipulation [14, 27, 28, 67, 86], but these typically train from large datasets depicting a single object. Comparison with existing tracking datasets is summarized in Table 1.

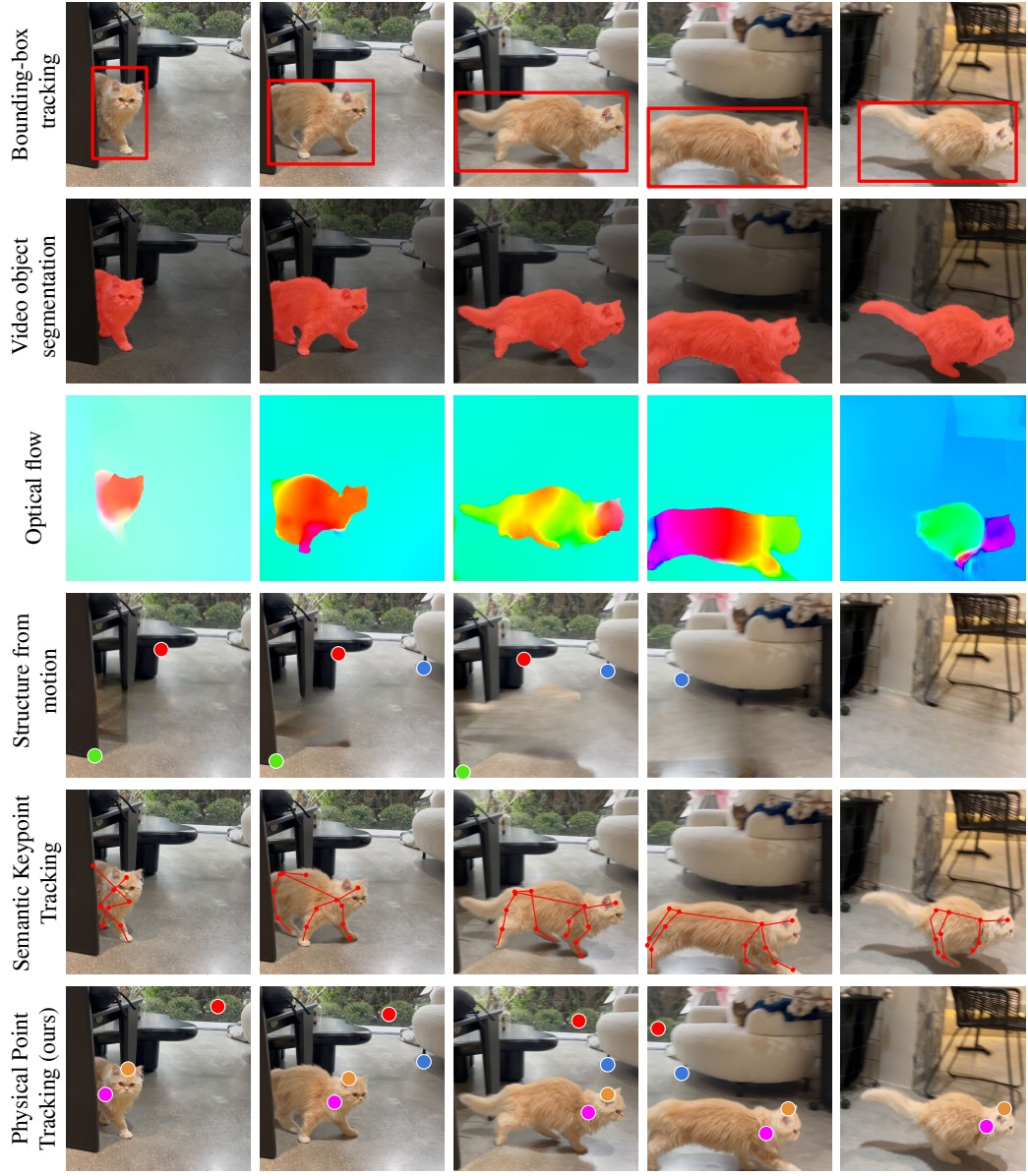

Figure 2: **Correspondence tasks in videos.** Most prior work on motion understanding has involved tracking (1) bounding boxes or (2) segments, which loses information about rotation and deformation; (3) optical flow which analyzes each frame pair in isolation; (4) structure-from-motion inspired physical keypoints which struggle with deformable objects, or (5) semantic keypoints which are chosen by hand for every object of interest. Our task, in contrast, is to Track Any Point on physical surfaces, including those on deformable objects, over an entire video.

One notable approach which combines several of the above paradigms is a concurrent work PIPs [20]. This work learns a form of long-term video point tracking inspired by Particle Videos [53], using a sim2real setup similar to TAP-Net. However, it struggles to find both training and testing data, relying principally on BADJA [5] for evaluation, which contains just 9 videos and tracks joints rather than surface points.

Our work is also related to "smart" software that aids human annotators, making guesses that annotators may accept or reject, so they don't need to label exhaustively. Polygon RNN [1] and UVO segmentation [77] accelerated image and video segmentation respectively, and similar systems have been proposed for object tracking in videos [32, 74, 75].

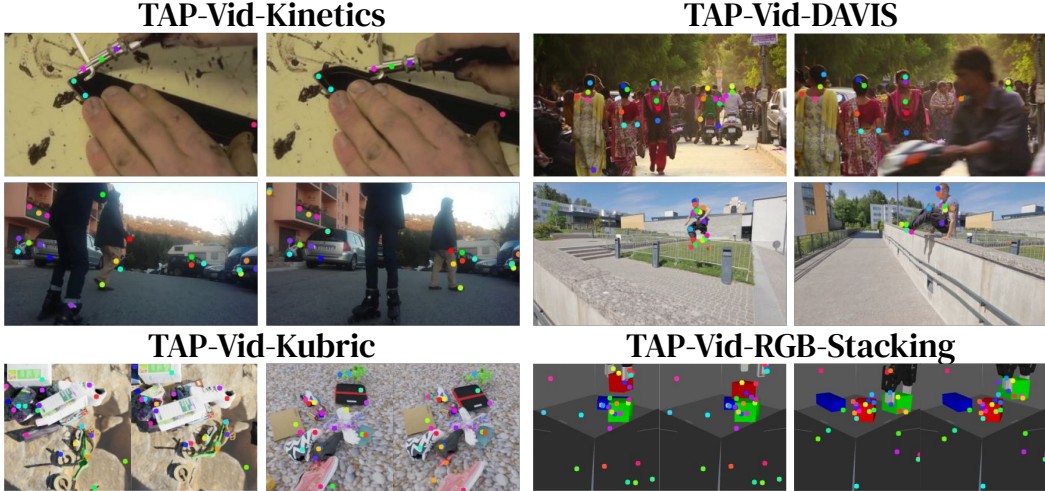

Figure 3: **The TAP-Vid point tracking datasets.** Ground-truth point annotations on two random videos from four point tracking datasets we use for evaluation—TAP-Vid-Kinetics and TAP-Vid-DAVIS containing real-world videos with point annotations collected from humans, the synthetic TAP-Vid-Kubric dataset, and TAP-Vid-RGB-Stacking from simulated robotics environment.

| Dataset | Type | #Videos (#Images) | Duration/ Time-scale | Long-term? | Point Precise? | Class Agnostic? | Non-rigid? |
|---|---|---|---|---|---|---|---|
| KITTI [16] | Optical flow | 156 | 400 frame pairs | ✗ | ✓ | ✓ | ✓ |
| COCO-DensePose [19] | Surface Points | (50k) | — | ✗ | ✓ | ✗ | ✓ |
| COCO-WholeBody [31] | Semantic Keypoints | (200k) | — | ✗ | ✓ | ✗ | ✓ |
| DAVIS [49] | Masks (multi-object) | 150 | 25 fps @ 2-5s | ✓ | ✗ | ✓ | ✓ |
| GOT-10k [24] | BBs (single-object) | 10k | 10 fps @ 15s | ✓ | ✗ | ✓ | ✓ |
| TAO [11] | BBs (multi-object) | 3k | 1 fps @ 37s | ✓ | ✗ | ✓ | ✓ |
| YouTube-BB [52] | BBs (single-object) | 240k | 1 fps @ 20s | ✓ | ✗ | ✗ | ✓ |
| PoseTrack [2] | Semantic Keypoints | 550 (37k) | train: 30 fps @ 1s eval: 7 fps @ 3–5s | ✓ | ✓ | ✗ | ✓ |
| 300VW [59] | Facial Keypoints | 300 | 30 fps @1–2 mins | ✓ | ✓ | ✗ | ✓ |
| ScanNet [10] | SfM 3D recons. | 1500 | ~1 min | ✓ | ✓ | ✓ | ✗ |
| MegaDepth [37] | SfM 3D recons. | 200 scenes (130k) | — | ✓ | ✓ | ✓ | ✗ |
| **TAP-Vid-Kinetics** | Arbitrary points | 1,189 | 25 fps @10s | ✓ | ✓ | ✓ | ✓ |

Table 1: **Comparison with tracking datasets.** Our dataset includes precise human annotated tracks of arbitrary class-agnostic points over long(-er) duration (10 seconds), unlike the existing datasets.

## 3  Dataset Overview

Figure 1 illustrates our general problem formulation. Algorithms receive both a video and a set of *query* points, i.e., a list of points $(x, y, t)$: $x, y$ for 2d position and $t$ for time. For each query point, the algorithm must output 1) a set of positions $(x_t, y_t)$, one per frame, estimating where that point has moved, and 2) a binary value $o_t$ indicating if the point is occluded on each frame. During occlusions, the $(x_t, y_t)$ output is typically assumed to be meaningless. Our benchmark combines both synthetic datasets—with perfect tracks but imperfect realism—and real datasets, where points must be tracked by hand. Note that with our real-world data, we principally aim for an *evaluation* benchmark. We expect TAP will be used in domains beyond the ones we annotate, e.g., novel robotics problems that have not yet been formulated. *Transfer* to evaluation datasets from synthetic data like Kubric is more likely to be representative of performance on unseen domains. This is thus the avenue we pursue in this work: we use Kubric for training the presented models, and hold out the other three datasets exclusively for testing. Note that evaluating the ability of an algorithm to track any point does not require us to label *every* point; the points must simply be a sufficiently random sample of all trackable points. Therefore, given a finite annotation budget, we prioritize diversity, and label a few points on each of a large set of videos.

| Dataset | # Videos | Avg. points | # Frames | Initial resolution | Sim/Real | Eval resolution |
|---|---|---|---|---|---|---|
| TAP-Vid-Kinetics | 1,189 | 26.3 | 250 | ≥720p | Real | 256x256 |
| TAP-Vid-DAVIS | 30 | 21.7 | 34-104 | 1080p | Real | 256x256 |
| TAP-Vid-Kubric | 38,325/799 | flexible | 24 | 256x256 | Sim | 256x256 |
| TAP-Vid-RGB-Stacking | 50 | 30 | 250 | 256x256 | Sim | 256x256 |

Table 2: **Statistics of our four TAP-Vid datasets**. Avg. points is average number of annotated points per video; # Frames is the number of frames per video. Note, the TAP-Vid-Kubric data loader can sample arbitrary points, and therefore there are functionally unlimited points per Kubric video.

For real-world evaluation, we annotate videos from the Kinetics-700 validation subset [7], a standard large-scale dataset for video understanding with diverse human actions, and the DAVIS validation set [49], a go-to benchmark for evaluating class-agnostic correspondence via segment tracking. For synthetic data, we turn to Kubric MOVi-E [18] and RGB-Stacking [34], as an example downstream task relevant to roboticists. For these synthetic datasets, we rely on the simulator to obtain ground-truth point tracks; details about the implementation can be found in the supplementary material. Table 2 lists basic statistics for our TAP-Vid benchmark, and Figure 3 shows example video frames. Our annotations are released under a Creative Commons Attribution 4.0 License.

## 3.1 TAP-Vid Datasets

**TAP-Vid-Kinetics.** This dataset consists of videos from the Kinetics-700 validation set [7], which represent a diverse set of actions relevant to people. We randomly sample videos which have 720p resolution available, as this aids annotation accuracy. These videos are unstructured YouTube clips, often with multiple moving objects, as well as moving or shaking cameras and non-ideal lighting conditions. Kinetics clips are 10 seconds, which we resample at 25 fps, resulting in 250-frame clips.

**TAP-Vid-DAVIS.** This dataset consists of videos from DAVIS 2017 validation set [49]: 30 videos intended to be challenging for segment tracking. We use the same annotation pipeline, but as DAVIS often contains only one salient object, we ask annotators to label up to 5 objects and 5 points per object. The point tracks were annotated at 1080p resolution to improve the precision. Like with TAP-Vid-Kinetics, the whole dataset is used for evaluation, resized to 256x256.

**TAP-Vid-Kubric.** We use the synthetic MOVi-E dataset introduced in Kubric [18] as both our main source of supervised training data and also for evaluation. Each video consists of roughly 20 objects dropped into a synthetic scene, with physics from Bullet [9] and raytraced rendering from Blender [8]. For training, we follow prior augmentation algorithms: cropping the video with an aspect ratio up to $2:1$ and as little as 30% of the pixels. A validation set of 799 videos from the same distribution is held out for testing.

**TAP-Vid-RGB-Stacking.** This synthetic dataset consists of videos recorded from a simulated robotic stacking environment introduced in [34]. We record 50 episodes in which teleoperators play with geometric shapes and are asked to display diverse and interesting motions. We use the *triplet-4* object set and the front-left camera. We sample 30 points per video from the first frame (20 on moving objects and 10 static objects/background) and track them via the simulator. The dataset is particularly challenging because objects are textureless and rotationally symmetric with frequent occlusions from the gripper, which are common in robotic manipulation environments.

## 4 Real-World Dataset Construction

Inspired by the Tracking Any Object (TAO) dataset [11], we aim for generality by allowing annotators to choose any object and any point they consider important, rather than specifying a closed-world set of points to annotate. Given a video, annotation proceeds in three stages, depicted in Figure 4. First, annotators choose objects, especially moving ones, without regard to their difficulty in tracking. Next, they choose points on each selected object and track them. Finally, we have a refinement phase where low-quality annotations are corrected by a different annotator, iterating as many times as needed. The work was performed by 15 annotators in Google's crowdsourcing pool, paid hourly. Note that, by using a relatively small and stable pool, annotators gain expertise with the tasks and the annotation interface in order to label most effectively. The full details of our design, including compensation rates, were reviewed by DeepMind's independent ethical review committee. All annotators provided informed consent prior to completing tasks and were reimbursed for their time.

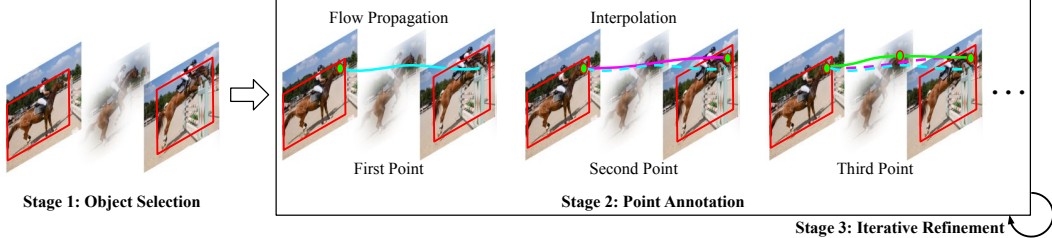

Stage 1: Object Selection | Stage 2: Point Annotation | Stage 3: Iterative Refinement

Flow Propagation | Interpolation

First Point | Second Point | Third Point

Figure 4: **Annotation workflow.** There are 3 stages: (1): object selection with bounding-boxes, (2) point annotation through optical-flow based assistance, and (3) iterative refinement and correction.

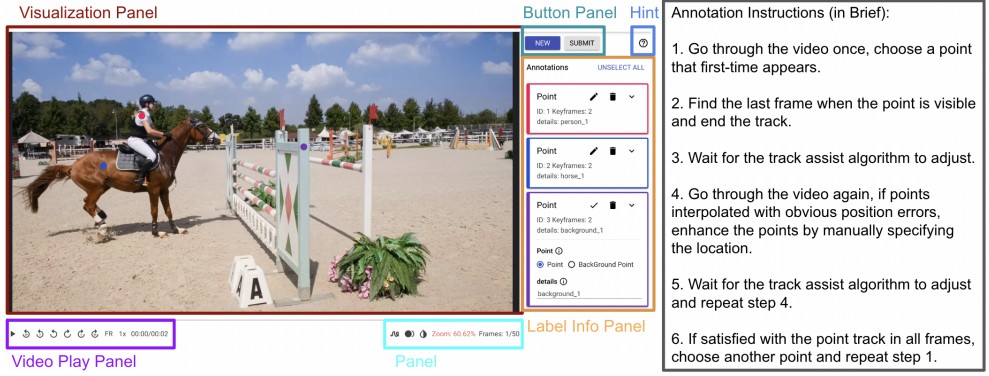

Figure 5: **Point annotation interface and instructions.** The interface consists of three components: visualization panel, buttons, and information panels. The instructions consist of six steps which guide annotators to iteratively add points with the help of the track assist algorithm.

**Stage 1: Object Selection** In the first stage, annotators choose up to $K$ objects ($K = 10$ for Kinetics and $K = 5$ for DAVIS). We ask annotators to prioritize objects that are salient and appear for a longer duration. We also prioritize moving objects and those producing sounds. We define an object as a standalone entity where it exists on its own, e.g. car, chair, person, which are solid, avoiding liquids/gasses and transparent objects (see supplementary for detailed instructions). Annotators then draw a box around each object once every 30 frames, using the interface from [32] (though we don't use any automated box tracking), and annotators also add a text label per object.

**Stage 2: Point Annotation** For every object selected in the first stage, annotators choose a set of $M$ points ($M = 3$ for Kinetics and $M = 5$ for DAVIS). Then they must annotate every other frame in the video with the corresponding location (except when occluded; annotators mark frames where points are not visible). Annotating every other frame makes annotation more accurate, as human eyes are tuned to whether or not a point is attached to a surface from how it moves. Exhaustively labeling every frame, however, is prohibitive, so we provide a novel *track assist* algorithm, which uses optical flow to convert sparsely-chosen points into dense tracks that follow the estimated motion. Annotators can choose as many or as few points as required to achieve the desired accuracy.

**Stage 3: Iterative Refinement** To ensure annotation quality, after initial submission, each annotated point goes to a second annotator, who checks and corrects points to achieve the desired accuracy. This iterative refinement continues until the last annotator agrees with all previous labels. On average, a 10s video takes approximately 3.3 hours to finish, often with 4–5 annotators working on the refinement.

## 4.1 Annotation Interface

Figure 5 shows the point annotation interface based on Kuznetsova *et al.* [32] presented to annotators in stages 2 and 3. The interface loads and visualizes video in the visualization panel. Buttons in the video play panel allow annotators to navigate frames during annotation. The information panel provides basic information, e.g., the current and total number of frames. The annotation buttons (NEW and SUBMIT) allow annotators to add new point tracks or submit the labels if finished. The label info panel shows each annotated point track and the associated 'tag' string.

Each track begins with an ENTER point, continues with MOVE points, and finishes when the annotator sets it as an EXIT point. Annotators can restart the track after an occlusion by adding another ENTER point and continuing. The cursor is cross shaped which allows annotators to localize more precisely. More details are provided in the supplementary.

## 4.2 Track Assist Algorithm

Given an initial point, modern optical flow algorithms like RAFT [66] can track a point somewhat reliably over a few frames, meaning they have the potential to annotate many frames with a single click. However, there are two problems: the estimates tend to drift if interpolated across many frames, and they cannot handle occlusions. Annotators manually end tracks at occlusions, but to deal with drift, we need an algorithm that allows annotators to *adjust* estimates created via optical flow to compensate for errors that would otherwise accumulate.

We first compute optical flow using RAFT for the entire video. When the annotator selects a starting point $p_s$ on frame $s$, we use flow to propagate that point from one frame to the next (using bilinear interpolation for fractional pixels) all the way to the last frame. When the annotator chooses a second point $p_t$ on frame $t$, we find the path which minimizes the squared discrepancy with the optical flow estimated for each frame. This corresponds to solving the following optimization problem:

$$\underset{\rho \in \mathcal{P}_{s:t}}{\arg\min} \sum_{i=s}^{t-1} \|(\rho_{i+1} - \rho_i) - \mathcal{F}(p_i)\|^2 \qquad \text{s.t.} \qquad \rho_s = p_s, \rho_t = p_t \qquad (1)$$

Here, $\mathcal{P}_{s:t}$ is the set of all possible paths from frame $s$ to frame $t$, where each path is described by a list of points. That is, each element $\rho$ in $\mathcal{P}_{s,t}$ is a list $\{\rho_i, i \in \{s, ..., t\}\}$ where $\rho_i \in \mathbb{Z}^2$. $\mathcal{F} : \mathbb{Z}^2 \rightarrow \mathbb{R}^2$ is the optical flow tensor, i.e., it maps image pixels to an optical flow vector.

While this is a non-convex optimization problem, it can be seen as a shortest-path problem on a graph where each node corresponds to a pixel in a given frame. Each pixel in frame $i$ is connected to every pixel in frame $i + 1$, with a weight proportional to the squared discrepancy with the optical flow at frame $i$. This can be solved efficiently with a form of Dijkstra's algorithm. In practice, the shortest path is typically found within a few seconds (faster than an annotator can click points) even when points are separated by dozens of frames (a few seconds of video). Annotators are instructed to check all returned tracks and to 'split' unsatisfactory interpolations, by adding more points that become new endpoints for the interpolation. If two points chosen by the annotators are closer than 5 frames apart, we assume the optical flow estimates are not helpful and interpolate linearly, and annotators have the option to fall back on linear interpolation at any time. We find that on average, annotators manually click 10.2 points per each track-segment (i.e., contiguous, un-occluded sub-track). There are about 57 points per segment on average, with a minimum of 2 manually-selected points per segment (the start and end, except for the occasional rare segments which are visible for a single frame). These results suggest that annotators are actively engaged in removing errors and drift from imperfect automatic propagation.

## 4.3 Evaluation and Metrics

We aim to provide a simple metric which captures accuracy at both predicting the locations of visible points and of predicting occluded points. However, comparing a binary decision (occlusion) with a regression problem isn't straightforward; therefore, similar to PCK [83], we threshold the regression problem so that it becomes binary. Then, as has been advocated for object box tracking [41], we compute a Jaccard-style metric which incorporates both position and occlusion.

Specifically, we adopt three evaluation metrics. (1) *Occlusion Accuracy (OA)* is a simple classification accuracy for the point occlusion prediction on each frame. (2) $< \delta^x$ evaluates the position accuracy only for frames where the point is visible. Like PCK [83], for a given threshold $\delta$, it measures the fraction of points that are within that threshold of their ground truth. While PCK typically scales the threshold relative to a body (e.g., a human), our metric simply assumes images are resized to 256x256 and measures in pixels. $< \delta^x_{avg}$ averages across 5 thresholds: 1,2,4,8, and 16 pixels. (3) The final metric, *Jaccard at $\delta$*, evaluates both occlusion and position accuracy. It's the fraction of 'true positives', i.e., points within the threshold of any visible ground truth points, divided by 'true positives' plus 'false positives' (points that are predicted visible, but the ground truth is either occluded or farther than the threshold) plus 'false negatives' (groundtruth visible points that are predicted as occluded or the prediction is farther than the threshold). *Average Jaccard (AJ)* averages Jaccard across the same thresholds as $< \delta^x_{avg}$.

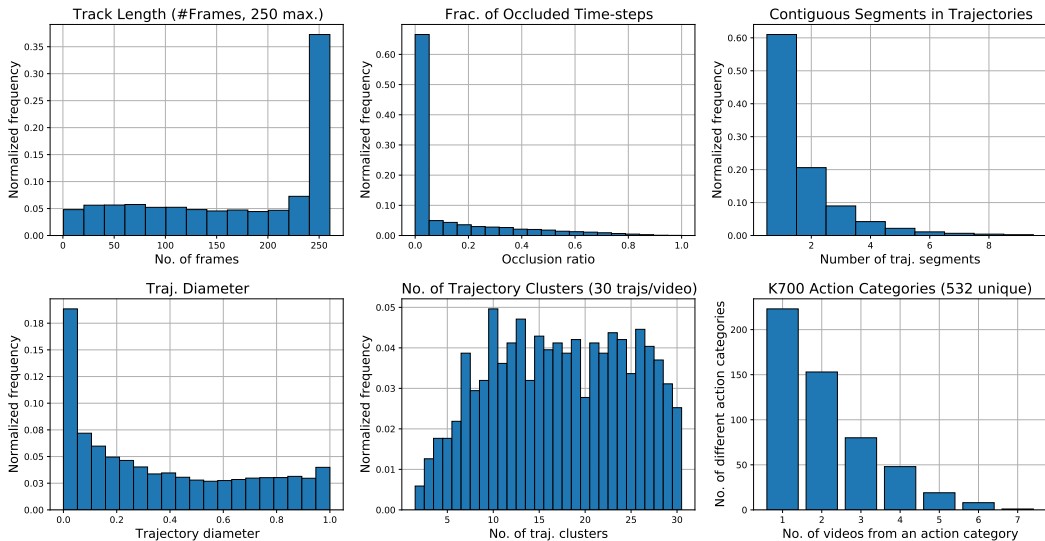

Figure 6: **Statistics of trajectories in TAP-Vid-Kinetics.** *Diameter* refers to the maximum distance between the positions of a point over time. *Trajectory Segments* refer to the different number of contiguous sections of point trajectories with breaks due to occlusion. *Trajectory Clusters* are formed from clustering the 30 trajectories per video with the distance between two trajectories being invariant to their relative offset; a large number of clusters indicate diverse point motion.

When evaluating on TAP-Vid, users are encouraged to train only on TAP-Vid-Kubric, but validation is a challenge, as TAP-Vid-Kubric performance may not correlate very well with performance on other datasets due to domain overfitting, especially with larger models. Therefore, if running a large number of experiments, we encourage users to use TAP-Vid-DAVIS as a validation dataset and hold out the other datasets purely for testing. Automated validation (e.g. large-scale architecture search) on the held-out datasets, even TAP-Vid-DAVIS, is discouraged.

## 5 Dataset Analysis

We first provide some basic dataset statistics. Then we validate the accuracy of our point tracking procedure, both qualitatively and manually annotating tracks on synthetic data where they may be compared with ground truth. Finally, to aid future research, we evaluate several baseline algorithms from the literature. Finding that they perform relatively poorly, we develop our own somewhat stronger baseline (TAPNet) using a version of cost volumes inspired from optical flow algorithms.

### 5.1 Point trajectory statistics

Figure 6 summarizes a few key statistics of the point trajectories in our large-scale TAP-Vid-Kinetics dataset. Most tracks span the entire duration of the clip (250 frames), are not occluded (55%), and are broken into a small number of segments due to occlusion ($> 70\%$ have $\leq 2$ segments). The point motion is often large (i.e., large trajectory diameter). To measure diversity, we cluster using agglomerative clustering (measuring trajectory distance via mean-centered distance between non-occluded points and stopping at 2 pixels; see supplementary for details), and find $\geq 85\%$ videos have $\geq 5$ clusters, suggesting diverse motion. Finally, a large number of Kinetics-700 action categories (532 unique) are represented in the videos.

### 5.2 Evaluation of human annotation quality

Our supplementary material contains qualitative examples of TAP-Vid-DAVIS. Overall, we observe that the tracks are quite accurate, as if the points are attached to the object. We also try without optical-flow-based track assistance (i.e. linear interpolation between chosen points); the resulting tracks drift visibly and do not look like they are attached.

Quantitatively, we adopt two approaches to evaluate the human annotation: 1) using simulated groundtruth, and 2) using human inter-rater reliability on real videos.

**Simulated Groundtruth** Kubric [18] contains all the information required to obtain perfect ground truth tracks for any point the annotators might choose. Therefore, we ask them to annotate 10 Kubric videos with length 2 seconds (50 frames at 25 FPS), where we have known ground truth point tracks. With our optical flow track assistance, over 99% of annotated points are accurate to within 8 pixels, 96% to within 4 pixels, and 87% to within 2 pixels. This is true even though Kubric includes many difficult objects that fall and bounce in unpredictable ways. In terms of annotation time, optical flow assistance improve annotation speed by 28%, from 50 minutes per video to 36 minutes per video. This is nontrivial considering the overall video length is only 2 seconds. See supplementary for more details, including a demonstration of the non-trivial improvement from our track assist algorithm.

**Human Agreement** After a first round of annotation on DAVIS, we perform a second round of annotation using the first frame points from the first round. That is, we ask different human raters to annotate the same set of points. We then compare the similarity between the human annotations on the same point track using our established metrics. Overall human agrees with 95.5% on occlusion and 92.5% on location with a 4 pixel threshold. On average, there is only a 1.46 pixel difference between two independent human tracks on the same point under 256x256 resolution. See supplementary for more details.

## 5.3 Baselines

Few existing baselines can be applied directly to our task. Therefore, we adapt several state-of-the-art methods which perform different kinds of point tracking with simple extensions. We expect that ideas from these papers can be better integrated into a more complete point tracking setup, but as baselines, we aim to keep the implementations simple. **Kubric-VFS-Like** [18]: A baseline for point tracking trained on Kubric using contrastive learning, inspired by VFS [81]. **RAFT** [66]: We extend this state-of-the-art optical flow algorithm to multiple frames by integrating the flow from the query point: i.e., we use bilinear interpolation of the flow to update the query point, move to the next frame, and repeat. We handle out-of-bounds points by using the flow from the nearest pixel, and mark points as occluded when they're outside the frame. **COTR** [30]: This method is intended to be used with larger baselines, so to propagate a query point, we apply this algorithm with the query frame paired with every other frame. We use cycle consistency inspired by Kubric-VFS-Like [18] to estimate occlusion. See supplementary for details.

## 5.4 TAP-Net

Overall, we find the above baselines give unsatisfactory performance. Therefore, we propose the first end-to-end deep learning algorithm to track any point in a video, aiming for simplicity. Our approach is inspired by cost volumes [23, 54, 85], which have proven successful for optical flow. We first compute a dense feature grid for the video, and then compare the features for the query point with the features everywhere else in the video. Then, given the set of comparisons between a query and another frame, we apply a small neural network which regresses to the point location (trained via Huber loss) and classifies occlusion (trained via cross entropy). See supplementary for details.

## 5.5 Results

Table 3 shows the comparison results. TAP-Net outperforms prior baseline methods on all 4 datasets by a large margin, and provides competitive performance with one concurrent work [20]. Kubric-VFS-Like provides somewhat competitive performance, especially on Kubric itself, but its cycle-consistency-based method for detecting occlusions is not particularly effective on real-world data, and the lack of end-to-end training means that its tracks are not very precise. COTR shares similar difficulties on detecting occlusions as it has no built-in method for doing so. Even ignoring occlusion estimation, COTR struggles with moving objects, as it was trained on rigid scenes. We see especially poor performance on the RGB-Stacking dataset; most likely the lack of textures results in degenerate solutions. RAFT likewise cannot easily detect occlusions, and due to the frame-by-frame nature of tracking, errors tend to accumulate, leading to poor performance. Concurrent work PIPs [20], however, provides perhaps the strongest competition, as it was designed for point tracking in video in the same spirit as our benchmark. It provides the best performance on TAP-Vid-DAVIS, although it struggles on TAP-Vid-Kinetics and TAP-Vid-RGB-Stacking. This is likely because PIPs is essentially an online algorithm; given the tracking result for $N$ frames, it performs inference on frames $N$ to $N + 8$ assuming that the result on the $N$-th frame is approximately correct, and performs only local searches. If the track is lost for more than 8 frames (e.g. a long occlusion) or if there is a discontinuity

| Method | Kinetics | | | Kubric | | | DAVIS | | | RGB-Stacking | | |
|---|---|---|---|---|---|---|---|---|---|---|---|---|
| | AJ | $< \delta^x_{avg}$ | OA | AJ | $< \delta^x_{avg}$ | OA | AJ | $< \delta^x_{avg}$ | OA | AJ | $< \delta^x_{avg}$ | OA |
| COTR [30] | 19.0* | 38.8* | 57.4* | 40.1 | 60.7 | 78.55 | 35.4 | 51.3 | 80.2 | 6.8 | 13.5 | 79.1 |
| Kubric-VFS-Like [18] | 40.5 | 59.0 | 80.0 | 51.9 | 69.8 | 84.6 | 33.1 | 48.5 | 79.4 | 57.9 | 72.6 | **91.9** |
| RAFT [66] | 34.5 | 52.5 | 79.7 | 41.2 | 58.2 | 86.4 | 30.0 | 46.3 | 79.6 | 44.0 | 58.6 | 90.4 |
| PIPs [20] | 32.8* | 53.6* | 74.7* | 59.1 | 74.8 | 88.6 | **42.0** | **59.4** | 82.1 | 37.3 | 51.0 | 91.6 |
| TAP-Net | **46.6** | **60.9** | **85.0** | **65.4** | **77.7** | **93.0** | 38.4 | 53.1 | **82.3** | 59.9 | **72.8** | 90.4 |

Table 3: **Comparison of TAP-Net versus several established methods on TAP-Vid**. TAP-Net outperforms all prior works–often by a wide margin–due to its end-to-end training and design choices. For starred entries, the underlying algorithm was not fast enough to run on the full Kinetics dataset in a practical timeframe, so we ran on a random subset. See supplementary for details.

in the sequence (e.g. a cut in the video), then PIPs is likely to fail catastrophically. DAVIS videos tend to be continuous shots with smooth motion, which means that the 'local search' and smoothness inductive biases are helpful; in Kinetics, however, there are more occlusions, cuts, and rapid motion like camera shake, which means that these inductive biases may be more harmful than helpful. PIPs also struggles on TAP-Vid-RGB-Stacking, possibly because it also contains long occlusions, but also possibly due to the textureless objects, a problem which might be attributed to the training dataset. PIPs is trained on FlyingThings3D [46], and it remains unclear if this is comparable to Kubric for training point tracking. See supplementary material for qualitative examples of tracking results.

## 5.6 Comparison to JHMDB

Although JHMDB is designed for human pose tracking [29], numerous prior works [26, 33, 35, 78, 81] use it as an evaluation for class-agnostic point tracking. The task is to track 15 human joints given only the joint positions in the first frame. This is an ill-posed problem, as the joints are *inside* the object, at some depth which is not known to the algorithm; worse, annotators estimate joint locations even when they are occluded. Despite the dataset's flaws, it remains the best available for point tracking on deformable objects, which explains its popularity (and underscores the need for something better). We demonstrate the lack of generality of JHMDB by showing that TAP-Vid can be used to improve performance on JHMDB, while the reverse doesn't hold. Specifically, we begin with a network pretrained on Kubric, and then fine-tune it using JHMDB for 5000 steps (see supplementary for details). Evaluating on TAP-Vid-Kinetics zero-shot, we obtain $36.4$ AJ, versus $46.6$ with no fine-tuning. Then, we took the same network pretrained on Kubric, and instead fine-tuned it on TAP-Vid-Kinetics (note that we don't typically expect users to train on TAP-Vid-Kinetics; this is merely a diagnostic for the dataset). We obtain $63.4$ PCK@0.1 on JHMDB, versus $62.3$ without fine-tuning. Interestingly, both numbers on JHMDB exceed the SOTA for class-agnostic tracking, $60.9$ from VFS [81]; our labels on Kinetics are good enough to improve performance over synthetic data alone. JHMDB, however, is too focused on a single class, and therefore harms performance under the same setting, demonstrating the inadequacy of prior semantic keypoint tracking when the goal is to track any point.

## 6 Conclusions

In this paper we introduce the problem of *Tracking Any Point* (TAP) in a given video, as well as the TAP-Vid dataset to spur progress in this under-studied domain. By training on synthetic data, a straightforward network TAP-Net performs better on our benchmark than prior methods. TAP still has limitations: for instance, we cannot handle liquids or transparent objects, and for real data, annotators cannot be perfect, as they are limited to textured points and even then may make occasional errors due to carelessness. We believe that the ethical concerns of our dataset are minimal; however, our real data comes from existing public sources, meaning PII and biases must be treated with care to ensure no harms to anyone in those sources, as well as fairness of the final algorithm. For our largest dataset, Kinetics, we will follow its approach of removing videos that are taken down from YouTube to ensure consent and issues with PII. Kinetics has also been vetted for biases [7]; training synthetic data should further minimize biases that might be learned by the algorithm. Visual tracking is a core capability of the human visual system [17, 61, 65, 68, 69]; widespread mechanization therefore risks amplification of malicious human actions. At the same time, advancements in TAP will potentially bring solutions to many interesting challenges, e.g., better handling of dynamic or deformable objects in SFM [56], and allowing the semantic keypoint-based methods [15, 42, 43, 72] often employed in robotic object manipulation to be applied to generic objects, independent of their class.

# 7 Acknowledgements

We wish to thank Klaus Greff and Andrea Tagliasacchi for discussions on Kubric; Alina Kuznetsova, Yiwen Luo, Aakrati Talati and Lily Pagan for helping us with the human annotation tool; Muqthar Mohammad, Mahesh Maddinala, Vijay Vibha Tumala, Yilin Gao, and Shivamohan Garlapati for help with organizing the human annotation effort; Adam Harley for help with PIPs; Jon Scholz, Mel Vecerik, Junlin Zhang, Karel Lenc, Viorica Patraucean, Alex Frechette and Dima Damen for helpful discussions, advice, and side projects that shaped this paper, and finally Nando de Freitas for continuously supporting the research effort.

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
