# OpenReview forum: "TAP-Vid: A Benchmark for Tracking Any Point in a Video"
_NeurIPS.cc/2022/Track/Datasets_and_Benchmarks — NeurIPS 2022 Datasets and Benchmarks _

### Official Review · Reviewer_YqcD · 2022-07-09
**Review on TAP-Vid**

**Rating:** 9
**Confidence:** 4
**Clarity:** Clear.

**Strengths:**

[Strongly motivated, practical, and novel task definition] While robust long-term tracking of any points from a video is an important problem for robotics and 3D vision tasks, there exists no notable benchmark specialized for this task. This task might be able to open a new video tracking task beyond visual tracking and video object segmentation.

[Clarity in writing and strong completeness] The writing is clear. The task is well motivated. The statistics and analysis on the dataset are compact, informative, and well summarized. Every claim is logically well justified. The process to annotate the labels is detailed, convincing, and practical. The suggested baseline method is reasonable, and cross-dataset evaluation highlights the importance of the new dataset.


**Weaknesses:**

For the final touch, it would be nice if the authors can provide an evaluation code, and explicit train/test/validation split.


**Additional Feedback:**

-

**Correctness:**

Overall, it is sound. However, the authors should be able to clarify the difference between TAP and matching (not optical flow).

**Documentation:**

Sufficient.

**Ethics:**

No.

**Relation To Prior Work:**

Clear.

**Summary And Contributions:**

This paper introduces a new benchmark dataset for the task of tracking any point (TAP), i.e., tracking an arbitrary 2D point defined at the first frame over the entire video. The dataset is composed of a mixture of real and synthetic data where for real data, human manually annotates the ground truth for a number of 2D salient points, and for synthetic data, perfect ground truth labels are provided. Analysis and statistics on the dataset, a new baseline model which outperforms existing methdos, and a comparison with existing datasets are provided.

---

> ### Author Response · Authors · 2022-08-10
> **Author response: Review on TAP-Vid**
>
> Thank you for your very positive review!
>
> “Matching” is a broad term which includes many different kinds of correspondence.  We assume that you are referring to keypoint matching for structure from motion, which is typically evaluated with datasets like HPatches.  From a problem definition perspective, these are largely the same: given a point in one image, the network should find a point on the same physical surface in another image.  However, despite conceptual similarities, existing datasets are quite different from ours, as prior works use rigid objects and don’t work with video, which removes useful cues like temporal continuity.  We have clarified in the introduction.
>
> We have added a train/val/test split (40% train, 20% val, 40% test, selected randomly) for the Kinetics dataset to our dataset zip file as others have also expressed interest in splits for this dataset.  However, we will still encourage sim-to-real transfer.  We also believe DAVIS is likely too small for training, and RGB-Stacking has strong biases that may be exploited by an algorithm, so we’ll allocate those entirely for evaluation.  We will provide evaluation code along with training code for TAP-Net; this is now undergoing cleanup and should be posted shortly.

---

### Official Review · Reviewer_QtD6 · 2022-07-12
**Novel Dataset with Flawed Execution**

**Rating:** 5
**Confidence:** 4

**Strengths:**

S1: TAP is a new, well formulated task that is clearly differentiated from tasks such as SFM, semantic keypoint tracking, and keypoint discovery.

S2: The track assist algorithm is novel, well evaluated, and useful for reducing annotation cost.

S3: The dataset collection process and content of the dataset is clearly described.

**Weaknesses:**

W1: How the dataset is utilized for evaluations is not clearly defined. Only TAP-Vid-Kubric has any mentions of a test split or the split’s size. None of the details in checklist item 3b are provided. Baseline methods lack any of the details described in checklist item 3d.

W2: The averaging of position accuracy metrics is not well motivated, with actual applications of these models likely to require accuracy at a domain-dependent fixed threshold. Averaging obscures whether there are specific thresholds where baseline models outperform TAP-Net.

W3: Discussion of impacts of the work are limited to technical impacts, with no discussion of positive or negative social impacts aside from the work of the Kinetics authors. **Clearly the NeurIPS ethics review guidelines were not adequately considered.**

W4: Use of existing assets is glossed over, not adequately answering checklist items 4b, 4d, 4e.

W5: Inadequate description of crowdsourcing participant compensation and potential risks presented to or found by Deepmind’s internal ethics committee.


**Additional Feedback:**

I would strongly recommend adding justifications to the answers given in the author checklist.

**Clarity:**

Writing is overall clear and concise. Figures and tables are well-made and unambiguous. However, there are notable absences in what was chosen to write about (W1, W3, W4, W5).

**Correctness:**

The dataset is constructed in a sound way. The evaluation of human annotations is clear and supports the claims made. The dataset statistics presented in Section 5.1 were a bit ambiguous. It appears that these statistics should more clearly be stated to only apply to TAP-Vid-Kinetics, as the first sentence suggests that the following statistics apply to the entire dataset.

As described in W1, the evaluation methods are ambiguous. It is unclear how training, validation, and test sets are created for each experiment. Model hyperparameters and tuning methodology are not described. Compute time and resources used for baseline model training appear to be missing.

The ambiguity of the evaluation methodology causes a lack of context for the results. As described in W2, the metrics also provide little actionable insight, aside from occlusion accuracy. An average across all delta values would provide a better measure of performance independent of threshold choice. Evaluations at specific thresholds would provide more actionable insight, even if only summarized in the main paper with tables relegated to the supplement.


**Documentation:**

Dataset collection is well documented, but tools utilized for collection are not shared. Dataset organization is not sufficiently detailed (W1). Availability and maintenance plans are not included or linked to in the paper, but a link in the supplement provides sufficient plans. Ethical and responsible use are not addressed in the paper, beyond the technical limitations of the paper (W3). There is neither sufficient detail nor code to support reproducible experiments (W1).

**Ethics:**

It is unclear whether the authors have consent to share the data and are complying with existing licenses (W4). Details of the treatment of data collectors, particularly their compensation, are not included (W5). No discussion of positive or negative social impacts aside from the work of the Kinetics authors is provided, showing the ethics review guidelines were not considered (W3). If data collectors were inequitably compensated, I would consider this a significant negative social impact.

**Relation To Prior Work:**

The TAP task is related to prior tasks such as SFM, keypoint tracking, and keypoint discovery. TAP differentiates itself with explicit modeling of occlusion and no restrictions on the points that should be able to be tracked.

No previous dataset provides long-term tracks, precise points, class agnosticism, and non-rigid objects at the same time. Previous works have utilized algorithms to assist in label generation in segmentation and object tracking, but none have been utilized for point tracking.

The models benchmarked on the dataset seem to be straightforward extensions of existing techniques.


**Summary And Contributions:**

This paper formulates the problem of “tracking any point” on video data. This formulation differs from related problems by aiming to track points starting from any pixel in any frame, not using a variation of keypoints to limit point selection. Additionally, predicting occlusion of the keypoints is used to avoid the complications of attempting to localize obscured points.

The dataset is comprised of four sources of data, two existing real world video datasets and two existing synthetic video datasets. Real world datasets are constructed by crowdsourcing bounding boxes and rough point tracking. More detailed point tracking is approximated with a novel use of optical flow estimates that are iteratively finetuned by annotators. Synthetic datasets utilize the simulators for ground-truth point tracks. The quality of crowdsourced annotations is measured by crowdsourcing annotations of this synthetic data and comparing point tracks.

Several existing algorithms are modified for use as baselines in the TAP task. An additional deep learning algorithm, TAP-Net, is proposed and found to outperform the baselines on all datasets in terms of occlusion accuracy, average thresholded position accuracy, and average Jaccard thresholded position accuracy.

---

> ### Author Response · Authors · 2022-08-10
> **Author response: Novel Dataset with Flawed Execution**
>
> Thank you for your careful review and for raising these considerations.
>
> **W1: Evaluation Methodology / Dataset Splits.** DAVIS, RGB-Stacking, and Kinetics are used exclusively for evaluation; we believed that the opening of section 3 and appendix 7.5 made this clear, but we realize now that we didn’t say this explicitly.  We made the decision to train exclusively on synthetic data as synthetic data makes it easier to control dataset biases, and furthermore, we believe that cross-dataset transfer will be more representative of real-world performance.  It is also in-line with optical flow evaluations, where the limited real data (KITTI) is typically held out from training.  However, as multiple reviewers have asked about data splits, we will also include a train/val/test split for Kinetics, though we will still encourage researchers to train on synthetic data.
>
> **W1: Other Training Details and Resource Utilization.** Regarding the other aspects of 3b (training details), for full reproducibility we will be releasing training code which reproduces our results; we believe there is sufficient detail for reproduction, including hyperparameters.  We have added a paragraph adding more details about hyperparameter tuning to our appendix section 7.5; we apologize for the original omission.  Regarding 3d, we thought it was clear that appendix 7.5 said we use TPUv3 using internal DeepMind resources.  We have improved our discussion of overall compute time in appendix 7.5.
>
> **W2: Averaging Metrics.** We agree that different applications may require different tracking precision.  Our appendix includes Table 2 with a breakdown across thresholds, and we will encourage researchers to include similar breakdowns (we couldn’t include these in the main text due to space constraints) However, we also note that datasets typically try to distill performance down to a single number in order to compare methods.  Our averaging strategy is inspired by COCO, a mainstay of computer vision, where object detection and instance segmentation both average the AP statistics across different thresholds.  We are unclear what you mean by “An average across all delta values would provide a better measure of performance” – this is exactly what we intended to provide.
>
> **W3: Ethical Considerations.** We as a team, and DeepMind as an organization, take ethics very seriously.  At submission, we believed that ethics considerations were adequately described in the main text, but we agree that more careful description in the checklist justifications will help the community understand our position.
>
> **W4: Licensing / Data Consent / PII / Offensive Content.** Regarding checklist items 4b, 4d, and 4e, we mentioned explicitly the license of the assets produced (RGB-Stacking and the points, which we own), and provided references to DAVIS, Kubric, and Kinetics, where the licenses are public information.  Regarding consent, the videos we’re using are either synthetic (no consent required) or already public, in which case we provide no further information about these videos than what can be inferred by visual inspection; any problems with consent here would also apply to the entire Kinetics dataset.  The annotators agreed to be compensated in return for providing data that can be used with machine learning models.  Regarding PII, DAVIS and Kinetics are public videos; we provide no PII that was not already public information.  Similarly for offensive content, Kinetics videos are from YouTube and must adhere to community guidelines for non-offensive content; DAVIS videos are manually curated as described by the dataset collectors.  We will include a further discussion of these details in the checklist, but if any of these details are inadequate, please tell us what remains unclear so that we can include them in the main paper.
>
> **W5: Annotator Compensation.** Google policy does not allow us to disclose the exact compensation structure for the annotators, which we agree is unfortunate, as we think it would assure you that our practices are fair.  DeepMind’s ethical committee has reviewed the entire structure of Google’s annotation procedures and found them adequate.  We are working with DeepMind’s ethics teams to find if more information can be disclosed, and will add information if we receive permission.

---

> ### Author Response · Authors · 2022-08-10
> **Author response pt 2: Novel Dataset with Flawed Execution**
>
> **W5: Risk Assessment.** We believe that the potential for social harm is small, a sentiment echoed by other reviewers.  Risks of bias are low as Kinetics has been vetted for bias.  We have considered many other risks and find no worse consequences than for any other machine learning technology.  For example, this could be used for surveillance, but arguably human tracking and human action recognition are worse in this respect, as nothing in our problem is tuned for people.  Inequitable use may arise to exacerbate inequality, but this is true of any technology which is controlled by its owner, and preventing this depends on social structures far beyond the scope of this paper.  Careless application can also lead to accidents, but this depends on downstream tasks, as point tracking is unlikely to be used on its own.  We have expanded our discussion of these issues, but we also believe that ethics discussions should be concise to ensure that people read them.  If you have specific potential harms that you think can arise from this work, please tell us and we would be glad to add them.
>
> **Dataset Statistics.** We have improved the discussion of 5.1 to clarify that these statistics refer to Kinetics.  We have also added justifications to all checklist entries.

---

> ### Author Response · Authors · 2022-08-28
> **Author Response: Review of Revisions**
>
> Thank you for your detailed insights.  We have made a few changes (mentioned below) and highlighted them in blue in the updated paper.  Note that QtD6 has made their review visible only to authors.  The interface makes it appear that replies to that post would also be visible only to authors (and not even to QtD6), so out of caution we are replying to the parent post.
>
> **W1: Evaluation Methodology / Dataset Splits:** We have updated line 85, adding a sentence to make it clear how the experiments are done.
>
> **W1: Other Training Details and Resource Utilization:** Indeed, we did not use any dataset-specific validation split.  As we don’t have a hidden test set, we are relying on users not to run an abnormally large number of experiments until they get a lucky model that would not be reproducible, which is common in computer vision (e.g. the ImageNet validation set, most optical flow datasets, etc).  Regarding hyper-parameter tuning, the little tuning we did used DAVIS performance as the main target metric; however, performance is far from perfect, so we don’t think overfitting is a concern.  When reporting scores for multiple datasets, we report performance for a single checkpoint across all datasets; there is no dataset-specific tuning.   However, we agree that our paper should propose a more standardized tuning procedure for future work which may use a larger number of experiments and have a larger risk of overfitting.  We have therefore added guidelines saying that DAVIS should be used for validation in section 4.3 of the main paper.  We are overall confident that TAP-Net can be reproduced.
>
> **W2: Averaging Metrics:** There are no thresholds where RAFT or Kubric-VFS-Like outperform TAP-Net.  Unfortunately, we no longer have access to the comparable numbers for COTR, but we believe the story was the same there.  We will perform an analysis including COTR and include metrics as requested.
>
> Indeed, performing an integral across all thresholds (logarithmically weighted by distance) is likely to be a slightly lower variance performance estimator than the ones we chose.  However, we found that variance was already low enough as we average across a large number of points.  Therefore, we opted for simplicity.  Again, this is common for evaluations in other domains, such as COCO and NYU Depth.  However, if variance turns out to be a problem in the future, we agree that an integral may be a useful strategy; it will not be difficult for other authors to add this as an extra metric given our dataset.
>
> **W3: Ethical Considerations:** We have added a small statement on potential abusive applications, although we are somewhat grasping at straws here, as we believe that these issues are rather generic and have been discussed better elsewhere.  [Guidelines for writing these ethics statements](https://medium.com/@GovAI/a-guide-to-writing-the-neurips-impact-statement-4293b723f832) say that it’s useful to focus on impacts that are “tractable, neglected, and significant”.  We believe that for datasets like this, bias is partially tractable, significant, and potentially slightly different in this work than in others, which is why we discussed our efforts to mitigate it.  For risks like potential amplification of malicious human actions, we believe that the ethical problems are already present in special-purpose systems that track face and body keypoints and object boxes.  We agree that ethics is important; again, if there are specific issues that you think are being missed, please tell us so we can add them.

---

### Official Review · Reviewer_QxLe · 2022-07-18
**A novel dataset for point tracking.**

**Rating:** 7
**Confidence:** 4
**Correctness:** Yes.
**Clarity:** Yes.

**Strengths:**

1. This paper constructs the first dataset for long-term arbitrary point tracking tasks. Such a dataset can contribute to the development of embodies agents.
2. Thorough comparison between the proposed TAP dataset and previous datasets for point tracking has been made in the paper. It helps a lot in figuring out the difference and contribution of the paper.
3. Comprehensive statistical information of the proposed dataset and the detailed annotation process has been provided.

**Weaknesses:**

1. Is there something like the error bar of the annotation provided? As demonstrated in the supplementary videos, some drifts or errors in the annotated points seem to exist, especially for non-rigid objects like the camel or black spawn. The drift or point disappearing happens even without occlusion.

2. Is the proposed TAP-Net trained on the combination of those sub-datasets or individually trained and tested on each sub-dataset? Suppose the model is individually trained on each sub-dataset. In that case, I am wondering about the generalization ability of the TAP-Net on the unseen sub-datasets. Such experiments can help better benchmark the model's generalization ability on unseen scenarios and make the dataset more important.


**Additional Feedback:**

Please see the Limitations part. I am looking forward to the authors' response and will increase my ratings if my concerns are well addressed.

**Documentation:**

Yes.

**Ethics:**

No.

**Relation To Prior Work:**

Yes.

**Summary And Contributions:**

This paper constructs a new dataset for arbitrary physical point tracking. The proposed task, i.e., tracking any points (TAP), can benefit the development of embodied agents. Although optical flow-based methods can perform well in short-term videos for arbitrary point tracking, the drift accumulation problem hinders their application in long-term scenarios. A novel baseline method TAP-Net is proposed to deal with the problem and gets superior performance on the proposed TAP-Vid series point tracking datasets.

---

> ### Author Response · Authors · 2022-08-10
> **Author response: A novel dataset for point tracking.**
>
> Thank you for your insightful review.
>
> For all experiments with TAP-Net, we train exclusively on Kubric, and hold out RGB-Stacking, DAVIS, and Kinetics purely for evaluation.  This is because we are interested in unseen domains (e.g. real-world robotic manipulation with novel grippers), and therefore, we expect cross-dataset transfer to be the best guarantee of generalization.  This also follows work in optical flow, where performance is often demonstrated via dataset transfer (e.g. flying chairs/flying things to Sintel or KITTI).
>
> Regarding accuracy, after a further review of the points, we agree that there are some errors that can be improved.  After submission, we gave feedback to our annotators and are in the process of another round of annotation.  We’ve already completed an updated version of DAVIS, which can be viewed [here](https://storage.googleapis.com/dm-tapnet/content/davis_ground_truth_v2.html).  We hope this addresses some of your concerns.  Unfortunately, while we agree that providing error bars would be useful, our annotation interface is somewhat legacy code that makes rather restrictive assumptions about data formats; it isn't possible to make, test, and deploy changes in the few weeks that we have for discussion, but we are exploring if this is possible in the longer-term.  We believe that the current annotations are still better than what can be achieved by any automated algorithm, so we want to make the dataset available immediately.  Reannotating the same points is a straightforward extension that we believe can be done in follow-up work if the community feels that the algorithmic performance is reaching a ceiling.

---

### Official Review · Reviewer_VmVz · 2022-07-27
**"tracking any point" (a good goal) turns into "watch all the failures in annotating hard to track points on deformable/articulated objects"**

**Rating:** 7
**Confidence:** 3

**Strengths:**

1. TAP-Vid's cross-dataset annotations allow for evaluation of generic motion tracking. Generic methods for motion tracking should work across datasets and domains. Since TAP annotations span multiple datasets, the benchmark is promising as a way to unify tasks and evaluations for general motion tracking in video. Existing dataset-specific evaluations may not be sufficiently expansive to evaluate generic motion tracking methods. This improves the significance of the contribution, as it paves the way for evaluating more generic methods.

2. TAP-Vid evaluating with points means it's a general benchmark for systems that might natively predict boxes/segments/etc. Point tracking naturally supports higher-level evaluations. Since TAP annotates point tracks, TAP evaluations can be run even on methods that only predict sprites/masks/canonical surface maps/other representations. This improves the relevance to the broader research community and accessibility by other researchers, as point tracking is a good interface for general tracking tasks regardless of method.

3. TAP-Vid builds on top of existing, valued, frequently-used datasets. Annotations of an existing dataset are useful in that they allow for more robust evaluations alongside tasks that are already valued. Since TAP provides annotations on Kinetics and DAVIS, it contributes to popular, in-use datasets. This increases the relevance to the broader research community, as the evaluation can be adopted to improve the evaluation richness of current methods.

4. TAP-Vid attempts to solve keypoints vs. joints. TAP-Vid rightly points out the conflicting definitions of keypoints depending on whether they refer to joints interior to a surface (as in human pose estimation) vs. whether they refer to surface points (in surface mapping and 3D tasks). A strength TAP-Vid is an attempt to move towards a clear, unified definition of point tracking while including the data and annotations that make such a new evaluation possible (allowing potentially for progress past JHMDB as shoehorned tracking testbed).

**Weaknesses:**

1. The major weakness of TAP-Vid is that annotation quality means "tracking any point" (a good goal) turns into "watch all the failures in annotating hard to track points". The major weakness here is the quality of the annotations. In 5/6 of the DAVIS videos I observed, there were deviations in a track from what should be the ground truth. This is a fixable problem, but the annotation quality is cornerstone to the validity and usefulness of the benchmark.

I looked at a few visualized RGB-stacking videos and the first six visualized DAVIS videos by index in the visualize.py script. Here are my notes from them:

DAVIS 0 – teal point, hair on the back of the goat – jumps around the goat significantly although clearly the point to track is the hair it was indexed on.
DAVIS 1 – orange point, side of the ladies face in the car – jumps to the front of the glasses before jumping back to the side of the head.
DAVIS 2 – green point, front of bike fender – jumps to below the bike and then above it.
DAVIS 3 – orange point, background ladies jeans – migrates to the elbow of the guy wearing the black sweater.
DAVIS 4 – lack of diversity of points, mostly positioned on the two ladies in the far background (not a quality issue)
DAVIS 5 – green dot, rear left quarterpanel BMW – migrates a foot or so along the car
DAVIS 6 – yellow dot, forehead right man – looks like the dot moved with the man and then back with a delayed reaction, moving it down below the ridge of his eyebrows briefly.
RGB-Stacking – these points are probably perfect due to their synthetic nature. It is interesting to notice points flicker in and out of occlusion.

To resolve this major weakness, the authors can do an annotation quality analysis in comparison to alternative keypoint annotations in video, like JHMDB (although they annotate joints). If they find the annotation quality is superior to JHMDB then this dataset is a net positive to the community. If the quality is lower, then this work risks introducing noise to learning to track keypoints over time. I currently think that Table 3 in the supplementary material is not enough to resolve this issue for me, as it is unclear whether the accuracy differences are do to predicting joints or predicting keypoints.

2. TAP-Vid creating "point-precise" annotations increases the stringency for "doing a good job". The “point-precise” nature of TAP and its annotations is a central motivator of the work, as expressed in Table 1. The difficulty here is that by choosing such a stringent task, it also significantly increases the quality expectations for succeeding at annotating this task. Given that a central argument is moving away from KEYpoint tracking to tracking any “trackable” points, it may be the case that annotators need more guidance on choosing trackable points, as they don’t seem to always succeed as is.

To resolve this weakness, resolve weakness 1 above. This weakness explains my perspective that there is self-imposed quality issues and to hint at where this issue might arise. It also offers an alternative resolution that is related to significant figures. If instead of "points" annotations are referred to as "blobs" and provided with spatial uncertainty estimates, it could reduce the stringency of discussing "points".

3. TAP-Vid does not ablate the value of concerning components in their assisted annotation suite. I am concerned that the annotation suite with track assist, an important part of the paper, may actually contribute to the poor quality. Although the supplement includes an analysis of the “improved accuracy” of using the optical flow track assist, it doesn’t include an ablation of the annotation suite. Specifically, the annotation instructions steps 4 and 5 and 6 seem like they always end with an invocation to the “track assist”. I am concerned that from the pseudocode there is no escape path that lets you “skip” the “track assist algorithm”. If true this means that an annotator doesn’t have the “final word” and always has to move on to a new point after the “track assist algorithm” has "refined" a point. It may be that the iterative refinement pseudocode is not included here.

To resolve this weakness, I would like clarity on the iterative refinement and whether or not every point is truly forced to be refined by "track assist". This weakness is also related to weakness 1 and could be resolved by improving/analyzing the quality of the data.

4. TAP-Vid's automatic tool is central to producing the annotations, yet due to how people respond to super accurate tools, the dataset isn’t really annotated in practice. Annotation speed is a potential cause for concern and source of quality problems. Annotation speed is important for the affordable creation of large-scale data. Assisting annotation for speed is valuable and done in this paper. From the paper, it seems that 2 annotator hours might go into tracking 30 points through every frame of a 25 FPS 10-second video clip. This seems to mean that 7500 (x,y,t) pairs will be annotated, at a rate of 62.5 annotations per minutes or roughly 1 annotation per second.
I am concerned that given the speed, the number of videos (over 1,100 videos from Kinetics) and the assist, it may be the case that an assisted tool that is correct 95% of the time is being rubber stamped by annotators. When automatic annotation tools become good enough, it can become the case that points are no longer explicitly checked and instead assumed to be correct.

To resolve this weakness, resolve weakness 1, etc. However, this weakness offers an alternative resolution -- avoiding or at least experimenting with annotation similarity with or without an annotation tool. If it turns out that a high percentage of no-tool-involved annotated point tracks are very similar to a sample of point tracks already annotated, it could similarly resolve accuracy concerns.

5. TAP-Vid's annotation instructions bias annotators towards certain object parts. The annotator guidelines bias annotators towards the example part tags “human hand, human head, car wheel”. This may be the reason behind the biased choice of points from DAVIS 4 above. The authors do not address or analyze the above bias that may be introduced in point selection by annotators. It may be the case that annotators only select certain points on certain objects, but this potential semantic bias is not discussed as the motivation of the paper is to annotate points regardless of their semantic class.

This is a minor potential/unexplored weakness in annotation quality related to semantic class and could be resolved by a semantic analysis of annotations.

6. TAP-Vid annotations may be systematically worse for deformable objects. Deformable objects are notoriously difficult to successfully track, yet they are an important part of any scene. Given the failure in DAVIS 0 to track the teal point on the back of the goat (and some of the other tracks), I am concerned that there are systematic annotation errors related to two axes: deformable and insufficiently distinct.

To resolve this major weakness (although the TAP authors stay away from semantic annotations), I would similarly (as in 5) recommend a semantic analysis of annotation quality. If it is the case that systematically points that are relatively pronounced within deformable/articulated objects are wrong, then it really hinders the value of this dataset for tracking. The authors do already analyze their annotator's quality on the Kubric MOVi-E dataset, I am unclear on the quantity of deformable/articulated objects that were present in this analysis (if any) and believe an analysis of point-track annotation accuracy conditioned on the semantics of objects would be immensely valuable.

**Additional Feedback:**

My primary concern is that "tracking any point" (a good goal) ends up turning into more of "watch all the failures in annotating hard to track points".  I like the concept but I'm worried about the execution. Any new data and annotations are good, but if new annotations are sufficiently noisy and low quality then no one will use them and they could harm tracking methods by providing false correct/incorrect signal.

**Clarity:**

The submission is clearly written and well-organized. When including the supplemental material it seems sufficient for one to replicate the results.

**Correctness:**

The submission is technically sound, though the annotation quality could use improvement. Claims are supported, as the collected tracks improve TAP-Net performance. The methods used are appropriate and the paper is a complete work, although the centrality of TAP-Net is a bit strange for a dataset submission.

**Documentation:**

Data collection, organization, availability, maintenance, and ethical and responsible use are all detailed well in the paper and on the project website. For the 3 metrics in the benchmark, there is a sufficient detail to support reproducibility.

**Ethics:**

Given the benchmark builds primarily upon prior datasets with new annotations, it raises few novel ethical concerns.

**Relation To Prior Work:**

The tasks and methods are similar to existing methodology. Keypoint tracking and optical flow estimation have a long, rich history. The provided related works accurately describe the similar research in the area.

Beyond the included related work, other works are slightly relevant. “Learning to Estimate Hidden Motions with Global Motion Aggregation” already attempts to improve upon occlusion in RAFT, and could be used as a replacement optical flow driver and baseline here. Further, recent works like “Deformable Sprites for Unsupervised Video Decomposition” accomplish similar correspondence tasks, but done with sprites instead of points.

**Summary And Contributions:**

TAP-Vid aims to expand keypoint tracking in video with a greater diversity of points and longer duration point tracks that account for occlusion. TAP-Vid introduces data, annotations, a benchmark, and a method for long-term tracking of points on solid objects in videos.

The contributed data is data recorded from a pre-existing robotic object-stacking simulator.
The contributed annotations are tracks of points on the surfaces of objects across four datasets: Kinetics, DAVIS, MOVi-E, and the RGB-Stacking robotic simulator.
The contributed benchmark consists of 3 evaluation metrics: occlusion accuracy (binary per point per frame occlusion), position accuracy (unscaled PCK), and a jaccard measure.
The contributed method produces features spatially for frames, selects the feature (or feature interpolations) that corresponds spatially and temporally to a given query point, then produces a matrix product between this feature and other features at the target time to produce a cost volume. This per-timestamp-pair per-qeury cost volume is then fed into a network that predicts a binary occlusion logit and a predicted (x,y) point that corresponds to the location of the query point in the target frame.

TAP-Vid is motivated by the data gap that exists between dense per-pixel tracking in short-term optical flow and sparse long-term keypoint tracking in video. The authors aim to fill in this gap with the above contributions.

TAP-Vid hypothesizes that if videos had more annotated point-tracks (including point-tracks that survive occlusion), then motion understanding in video would improve. TAP-Vid tests this claim by training a network, TAP-Net, that uses these new point-tracks. TAP-Net’s performance is generally improved when using these point-tracks vs. point-tracks from the existing JHMDB dataset. This analysis is slightly, but bearably, confounded by the differing definitions of keypoints that exist between human pose estimation and their surface point tracking.

---

> ### Author Response · Authors · 2022-08-10
> **Author response pt 1: "tracking any point" (a good goal) turns into ...**
>
> Thank you for your extremely detailed and careful review.
>
> **1, 2: Annotation Quality / DAVIS Points / Precision.** We acknowledge that the annotated points are imperfect; in practice, humans cannot make perfect judgements and don’t have the patience to independently label every frame.  However, we are not aware of any alternative method to track any point aside from human labeling, and so if we want to solve the problem, we must work with what we have.
>
> That said, we agree that we can do better; in fact, we had already begun another refinement stage before even receiving the reviews, which fixes or substantially improves almost all of the points that you mentioned.  We have posted an updated version of [our DAVIS points](https://storage.googleapis.com/dm-tapnet/content/davis_ground_truth_v2.html) which you can browse.  We are applying the same re-annotation to all of the Kinetics points, which we will complete before the camera-ready deadline.
>
> While we agree that spatial uncertainty estimates would be useful for validating our results and making our estimates more meaningful, our interface does not currently allow us to re-annotate the same point multiple times; we have requested this feature, but it is non-trivial to build.  However, we believe that this is straightforward as follow-up work that can be added if the community believes it is valuable.
>
> **1: Flickering Points in RGB-Stacking:** We also noticed this flicker during our rendering.  Recall that occlusion is determined by whether the point lies behind the depth map.  For RGB-Stacking, the scene depth was estimated by a simple nearest neighbors; aliasing on strong discontinuities can lead to unstable depth estimates and flickering.  We have rewritten this to follow Kubric’s approach (using the max of four nearest neighbors to determine depth) which has removed most of the flickering.
>
> **3 & 4: Annotator Engagement/Reliance on Track Assist.** The annotators do have the option to turn off the track assist algorithm, which will result in points being interpolated linearly; we’ve clarified this point in the paper.  In practice, we find much poorer quality without the track assist algorithm, as camera shake and object motion are pervasive in real videos.  Furthermore, annotators can alter any point produced by track assist, and track assist will never modify a point input by the annotators.  In this sense, the annotator always has the final say. We find that the annotators do not simply “rubber stamp” the track assist’s output, as the average number of manual point annotations per track segment (contiguous sub-track without occlusion) in the dataset is 5.8 points; as a comparison, in case of “rubber stamping” / without active annotator engagement, there would only ever be 2 points per segment corresponding to the first and the last points of the segment.
>
> Furthermore, note that for annotators reproducing the behavior of the automatic tool is clearly not enough, as there are instances of cuts and occlusions that RAFT cannot handle, and dealing with this (as well as drift) remains an open research problem.  We believe that this dataset already provides a gradient to research, and also a problem that would be very useful to us at DeepMind and many others if quantitative performance were improved.
>
>
> **4: Annotation Quality Analysis w.r.t. JHMDB.** It is unclear to us how we can compare accuracy to JHMDB, as we cannot find any references to how JHMDB evaluates the accuracy of its points.  It makes no attempt to annotate synthetic data where ground-truth joints are known; moreover, it annotates points which are not directly observable in the image, inferring their locations approximately by asking annotators to pose a 2D puppet over the image segmentation.   Typical evaluation thresholds are far looser than ours: e.g. PCK@0.1 is correct if the point is within 10% of the size of the person’s torso; by this measure, many of the points you point out as errors in our DAVIS evaluation would, in fact, be correct according to JHMDB’s metric (see Figure 2 in Dantone et al. “​​Human Pose Estimation using Body Parts Dependent Joint Regressors,” the source of JHMDB’s metrics).  These facts suggest that our ground truth is, in fact, more accurate than JHMDB, but there’s no way to be certain, as manual annotations are the only available source of ground truth for real images.

---

> ### Author Response · Authors · 2022-08-10
> **Author response pt 2: "tracking any point" (a good goal) turns into ...**
>
> **4: Ablate Track Assist for Quality Analysis.** Regarding comparing point tracks with and without track assist, we provide an experiment (section 5.2) where annotators label the same Kubric videos (with known perfect ground-truth) with and without track assist, and find that track assist gives results closer to ground truth.  In practice, however, we found that the average case with Kinetics videos was even worse: camera shake is pervasive in real-world videos, meaning that annotators might need to label every single frame in isolation.  Surprisingly, this gave poor results even when annotators took a long time: we speculate that it’s difficult to judge and fix drift over short timescales because our eyes cannot easily judge motion on still frames.  It is only after annotating many frames that annotators can play back their annotations and notice jitter, and they’re likely to give up quickly.  This problem nearly killed the project in its early stages, until we discovered (somewhat to our surprise) that point tracks were much better with the tool.  Based on this experience, we expect that tracks provided without the tool will actually be too poor in quality to be a meaningful comparison.
> Also, remember that for our evaluation metrics, images are resized to 256x256.  Therefore, errors are actually smaller than what’s visible on the higher-resolution videos shown on our website.  We only showed them at that resolution for clarity.  Furthermore, we don’t intend that the human labels should be used for training, only the perfect synthetic ground truth; therefore, there’s little risk that downstream algorithms will reproduce the errors in this dataset.
>
> **5 & 6: Semantic Analysis of Annotated Points.** Regarding a semantic analysis, we ask annotators to choose moving, foreground objects.  We agree that this is a bias, but we argue that it is a useful one, as moving objects are more difficult to track, more relevant to common human goals, and are not currently handled well by structure-from-motion datasets.  To better understand the semantic biases, we now include the full list of semantic labels added by annotators in our supplementary (section 10), along with a brief discussion.  We see a huge range of objects: >1400 unique labels, though there are some semantic duplicates).  Unsurprisingly, we do see a bias towards humans: 26.1% are on humans and 23.4% on clothing (on or off the body), but 50.4% are on other types of objects, including indoor and outdoor objects, vehicles, and animals.
>
> **6: Deformable Objects.** Unfortunately Kubric does not currently support deformable objects.  We have contacted the Kubric authors regarding whether these can be added, but unfortunately, it seems non-trivial effort will be required.  We do not believe that deformable objects are significantly more difficult for RAFT, as RAFT is tuned for Sintel, which contains deformable objects, and this is validated by the results on other animals in DAVIS.  Likely more points will need to be labeled for deformable objects, but we hope that the new version of DAVIS convinces you that this is possible.

---

> ### Comment · Reviewer_VmVz · 2022-08-23
> **Annotation quality improvements are sufficient to address my concerns**
>
> The authors have improved the annotation quality on DAVIS with further refinement, and claim they will do the same for Kinetics. Quality was my major concern, and because this has been addressed I am changing my review to an acceptance.
>
> I really appreciate that the authors individually addressed all the weaknesses I mentioned even though they were primarily riffs on the same concerns about quality. I hope that the discussion was valuable in conveying how different types of errors might be differently detrimental to quality.

---

### Official Review · Reviewer_mNB7 · 2022-07-28

**Rating:** 9
**Confidence:** 3
**Correctness:** Yes.
**Clarity:** Yes.

**Strengths:**

- The problem setting is interesting and very challenging for existing algorithms. It opens new research questions and I believe there will be a lot of follow-ups to improve the tracking performance on this benchmark.
- The annotations have high quality. I check the webpage and each subset has some sample videos (thank you!). The quality is beyond my expectation. They are very useful.
- The scale of the dataset is large. Given that each video clip is 10s, 532 real videos are a lot for annotation of the trajectory over all frames. Two synthetic subsets are also proposed to help the training.
- Experiments are extensive. Three types of baselines are proposed, as well as a novel approach TAP-Net.


**Weaknesses:**

I don't see any obvious weakness in this paper.

**Additional Feedback:**

N/A

**Documentation:**

Yes.

**Ethics:**

I can't foresee any issues. Videos come from Kinetics, DAVIS and renderings.

**Relation To Prior Work:**

Yes.

**Summary And Contributions:**

The paper proposed TAP-Vid dataset, which is a benchmark for tracking any point in a video. Given a video clip and a set of query points, the goal is to predict trajectories over the whole video whether it is occluded or not. The benchmark contains four parts of data which are basically sources of videos: Kinetics, DAVIS, Kubric and RGB-Stacking. Two of them are real videos and the others are synthetic. Synthetic videos provide a good training set, while real videos can be both useful in training and validation. For each video, 25 query points are sampled and the trajectories are manually annotated. Baselines have unsatisfactory performance on TAP-Vid dataset and suggests a lot of space to improve.

---

> ### Author Response · Authors · 2022-08-10
> **Author response: Review**
>
> Thanks for your very positive review!

---

### Official Review · Reviewer_xA9H · 2022-07-28
**Review for TAP-Vid**

**Rating:** 7
**Confidence:** 5
**Correctness:** The claims appear to be correct.
**Clarity:** The paper is well written.

**Strengths:**

1. The research field of video TAP is under-studied. This paper introduces a companion benchmark it would benefit the community.
2. The data collection method provided in this paper shows a standard pipeline for the following research.
3. The proposed baseline method presents the usability of the benchmark.

**Weaknesses:**

1. The method could not be used for transparent objects.
2. The synthetic data may not lead to bias in model evaluation.

**Additional Feedback:**

No additional feedback.

**Documentation:**

The details of data collection and organization are clear.

**Ethics:**

No ethical concerns.

**Relation To Prior Work:**

Prior works are compared with the proposed dataset and the difference in scale is highlighted.

**Summary And Contributions:**

In this paper, the authors propose a point tracking dataset with 1189 real videos. Three baseline methods are designed to evaluate the performance.

---

> ### Author Response · Authors · 2022-08-10
> **Author response: Review for TAP-Vid**
>
> Thank you for your positive review!  Indeed, transparent objects are currently out-of-scope, although we believe that extensions are possible by re-defining the task, for example, by specifying the target surface as the closest physical surface, or more abstractly by using language to define objects.  We also agree that synthetic data may have biases, which is why we encourage training on synthetic data (to guarantee accuracy of the training data), but evaluate on real data (to guarantee real-world performance).  We believe that overcoming the gap between synthetic and real data presents an open challenge to the community.

---

### Author Response · Authors · 2022-08-10
**Overall author response and updates**

We thank all of our reviewers for their efforts and helpful feedback. We have posted updated main and supplementary pdf files which address the concerns raised: changed text is highlighted in red.

We have also updated versions of the three zip files containing our dataset annotations/videos.  RGB-Stacking has been improved to remove flickering, which was due to aliasing of the depth map.  DAVIS has been improved via another round of refinement, a process which was started before we received reviews back.  We have also added splits to Kinetics; we are in the process of a further refinement of Kinetics following the procedure used in DAVIS.  We will re-run our experiments on the updated datasets.  The webpages for RGB-Stacking and DAVIS have been updated to show the new ground truth.  New URLs for the datasets are:

[DAVIS](https://storage.googleapis.com/dm-tapnet/tapvid_davis_f4349ed22a0be646f1813bdd532f585c.zip)

[Kinetics](https://storage.googleapis.com/dm-tapnet/tapvid_kinetics_f4349ed22a0be646f1813bdd532f585c.zip)

[RGB-Stacking](https://storage.googleapis.com/dm-tapnet/tapvid_rgb_stacking_f4349ed22a0be646f1813bdd532f585c.zip)

---

### Meta-Review · Area_Chair_4jAJ · 2022-09-09

**Recommendation:** Accept
**Confidence:** 4

**Metareview:**

TAP-Vid presents a benchmark that will be highly useful for tracking research for tracking arbitrary physical points on surfaces over long video clips. The reviews are all positive, with one reviewer raising ethical aspects in preparing the benchmark.  I find the rebuttal sufficient and adequate and hence recommend acceptance of the paper.

---

### Decision · Program_Chairs · 2022-09-16

Accept